# A Systematic Review of Companion Diagnostic Tests by Immunohistochemistry for the Screening of Alectinib-Treated Patients in ALK-Positive Non-Small Cell Lung Cancer

**DOI:** 10.3390/diagnostics12051297

**Published:** 2022-05-23

**Authors:** Sulim Kang, Jaehyun Woo, Sungmin Kim

**Affiliations:** 1Department of Medical Industry, Dongguk University-Seoul, 26, Pil-dong 3-ga, Jung-gu, Seoul 04620, Korea; slkang2047@gmail.com (S.K.); highhighambition@gmail.com (J.W.); 2Department of Medical Biotechnology, Dongguk University, Bio Medi Campus, 32, Dongguk-ro, Ilsandong-gu, Goyang-si 10326, Gyeonggi-do, Korea

**Keywords:** companion diagnostic test, immunohistochemistry, alectinib, ALK, non-small cell lung cancer

## Abstract

Companion diagnostic tests and targeted therapy changed the management of non-small cell lung cancer by diagnosing genetic modifications and enabling individualized treatment. The purpose of this systematic review is to assess the clinical applicability of companion diagnostic tests (IHC method) by comparing the effects of alectinib and crizotinib in patients with *ALK*-positive NSCLC. We searched for literature up to March 2022 in PubMed, Web of Science, Cochrane, and Google Scholar. The inclusion criteria were randomized controlled trials comparing the effectiveness of alectinib and crizotinib using an IHC-based companion diagnostic test. The primary outcome was progression-free survival (PFS). The secondary outcomes were objective response rate (ORR), duration of response (DOR), and overall survival (OS). PFS was longer in alectinib (68.4 [61.0, 75.9]) than crizotinib (48.7 [40.4, 56.9]). This indicated that alectinib had a superior efficacy to that of crizotinib (HR range 0.15–0.47). In all secondary outcomes, alectinib was better than crizotinib. Particularly for the ORR, the odds ratio (OR) confirmed that alectinib had a lower risk rate (OR: 2.21, [1.46–3.36], *p* = 0.0002, *I*^2^ = 39%). Therefore, the companion diagnostic test (immunohistochemistry) is an effective test to determine whether to administer alectinib to *ALK*-positive NSCLC patients.

## 1. Introduction

Non-small cell lung cancer (NSCLC) is cancer that constitutes over 80% of all lung cancers and is a major cause of mortality worldwide [1,2]. The survival rate of NSCLC is about 23% at 5 years and with advanced disease, the survival rate is about 6%. There is a tremendous need for the high-quality diagnosis and treatment of patients with NSCLC [3]. Several oncogenes are involved in the pathogenesis of NSCLC: epidermal growth factor receptor (*EGFR),* B-Raf *(BRAF),* anaplastic lymphoma receptor tyrosine kinase (*ALK),* MET proto-oncogene *(MET)*, and c-ros oncogene 1 *(ROS1*) [4]. In particular, the rearrangement or fusion of the anaplastic lymphoma receptor tyrosine kinase (*ALK*) gene was first identified in 2007 as a driver of lung cancer [5]. It is overexpressed in cancer in the form of translocations with other partner genes, leading to the fusion of oncogenes [6]. Although *ALK* rearrangement occurs in about 5% of NSCLC, it has recently been confirmed to have a powerful transforming effect in lung cancer patients [5,7]. For the treatment of *ALK*-positive NSCLC, four inhibitors have been developed over the past decade: crizotinib, ceritinib, alectinib, and brigatinib. Among them, alectinib is considered an optimal therapeutic drug because of its potential to have the best clinical efficacy and safety [8].

Two main methods can be used to diagnose mutations in NSCLC: fluorescence in situ hybridization (FISH) and immunohistochemistry (IHC). These are methods that are still widely used in clinical laboratories despite the recent development of molecular tests [9]. FISH is the reference standard test for diagnosing NSCLC and is the only validated method. However, there are limitations related to the required technical skill, high cost, and securing of sufficient tumor cells. IHC, which is an alternative to FISH, is being used more in clinical fields because of its excellent performance and cost-effective advantages [10,11,12]. Also, in clinical studies, IHC tests in low-volume biopsies and fine-needle aspirations (FNA) showed excellent results, proving to be a better method than FISH [13]. A device for screening *ALK*-positive NSCLC patients using the IHC method typically includes ‘VENTANA ALK (D5F3) CDx Assay (VENTANA Medical Systems, Inc., Oro Valley, AZ, USA)’, which is approved by the FDA. It qualitatively detects the *ALK* protein in formalin-fixed paraffin-embedded (FFPE) samples of NSCLC patients to be treated. In addition, VENTANA provides a good performance to enable seamless detection of *ALK* rearrangements [14].

Analytical methods may vary depending on the sample types. A recent paper provided diagnostic recommendations for both histology and cytology specimens [15]. Immunocytochemical (ICC) analysis of cytological samples is the only diagnostic method for inoperable NSCLC patients. The minimum number of tumor cells required for *ALK* ICC analysis is 200 as a cut-off. In one study, the correlation between *ALK* FISH and *ALK* ICC was strong (93.6%) and the negative predictive value of *ALK* ICC was high (95%). Due to the fact that *ALK* rearrangements are rare in lung cancer, most samples will be negative. Therefore, the high negative predictive value of *ALK* ICC is particularly useful for *ALK* detection because it can be used to screen most negative samples at a relatively high level of confidence [16]. ICC is increasing the demand for biomarker diagnostics in cytology specimens as minimally invasive diagnostic methods advance [17]. Recent methods include reverse transcription-polymerase chain reactions (RT-PCR) and next-generation sequencing (NGS). RT-PCR is a fast and easy assay to detect *ALK* using specific fusion primers. In addition, there is no subjective analysis bias and this applies to the cytological analysis of patients in which it is difficult to apply FFPE samples [18]. However, the need for complex ribonucleic acid (RNA) pretreatment may limit clinical applications. NGS can analyze a limited amount of tissue samples and can simultaneously identify multiple drivers of oncogenes [19]. Both assays have good sensitivity: IHC 83–100% (D5F3), 93–100% (5A4) and NGS 85–100% [20]. However, this requires a sample with a higher proportion of tumor cells and has limitations in that it takes a long time to obtain data (Table 1) [21].

Traditional NSCLC treatment includes chemotherapy, which applies the same treatment for the same disease. Conversely, treatment in precision medicine involves the selection of biomarker-based treatments including companion diagnostic tests [3]. The companion diagnostics (CDx) detect genetic biomarkers to diagnose whether the drug is effective in patients [22]. They understand the pathophysiology of tumors to enable the stratification of patients for the receipt of targeted drugs based on expected responses to drugs and potential prognosis [23]. Recently, drug development and targeted treatment have been undertaken as molecular gene therapy has become possible [24]. Over the next decade, companion diagnostic testing is expected to revolutionize drug discovery and development processes and personalize health care by customizing the selection of existing and novel therapeutics [25]. The identification of genetic modifications that enable individualized treatment is changing the management of NSCLC [26]. The accurate diagnosis of NSCLC is necessary for the development and application of these treatments. Therefore, the clinical effectiveness of companion diagnostic tests is critical to provide treatment strategies to NSCLC patients [27].

In Ocana et al., a systematic literature review and meta-analysis were performed on the effect of companion diagnosis on the safety and tolerance of targeted anticancer drugs. Targeted anticancer drugs using companion diagnosis showed a low toxicity rate and an improvement in PFS, confirming companion diagnoses as a viable option [28]. In Pyo et al., a systematic review and meta-analysis were conducted to investigate the diagnostic accuracy of *ALK* immunohistochemistry in NSCLC patients [29]. In this study, by analyzing the performance and concordance between FISH and IHC, the correlation was identified, and it was proved that IHC demonstrates a strong consensus with FISH and has a high diagnostic accuracy. Liu et al. evaluated the diagnostic accuracy and clinical value of IHC for detecting *ALK* [30]. The analysis confirmed that IHC has excellent sensitivity and specificity and can be a simple and fast method to detect *ALK*. However, to the best of our knowledge, no systematic review has been conducted on whether an IHC-based CDx test has clinical applicability to the selection of patients for targeted therapy.

The purpose of this systematic review is to assess the clinical applicability of a companion diagnostic test (IHC method) by comparing the effects of alectinib and crizotinib in patients with *ALK*-positive NSCLC.

## 2. Materials and Methods

The protocol of the study was reported according to the Preferred Reporting Items for Systematic review and Meta-Analyses (PRISMA) statement (PROSPERO, number CRD42022304862). The PRISMA Checklist is in the Appendix A.

### 2.1. Search Strategy and Criteria of Study Selection

We searched for literature up to March 2022 in PubMed, Web of Science (WoS), Cochrane Register of Controlled Trials, and Google Scholar. Keywords were “Companion diagnostic test” or “CDx” and “Non-Small Cell Lung Cancer” or “NSCLC” and “anaplastic lymphoma receptor tyrosine kinase” or “ALK” and “alectinib” and “crizotinib” and “Randomized clinical trial”. The inclusion criteria were randomized controlled trials (RCT) comparing the effectiveness of alectinib and crizotinib using an IHC-based companion diagnostic test. We included studies in *ALK*-positive NSCLC patients and studies in which at least one appropriate outcome was reported. The target patients were set as total NSCLC patients without the restriction of tissue subtype. Irrelevant literature, letters, reviews, and clinical trial protocol literature were excluded.

### 2.2. Data Extraction and Quality Assessment

Data were extracted as follows: study, year, study type, number of patients, age, intervention, comparison, CDx test, and outcomes. Two reviewers (*Kang* and *Woo*) independently extracted data. Any disagreement was resolved through discussion and joint review. The risk of bias was evaluated using the ‘Methodology Checklist 2014’ tool of SIGN, UK.

### 2.3. Data Analysis

In the ACCE (Analytical validity, Clinical validity, Clinical utility, Ethical/legal/social issues) model project, clinical utility is defined as “the possibility that a test will significantly improve patient outcomes” [31]. Outcomes were established, including results of treatment and drug response that could directly affect survival. The primary outcome was progression-free survival (PFS). If the median value was reported, it was extracted as the median PFS, and it was extracted including the hazard ratio value. The secondary outcomes were objective response rate (ORR), duration of response (DOR), and overall survival (OS), which are indicators of the drug response rate. In the case of DOR, if a median value was reported, it was extracted as the median DOR, and it was extracted including the hazard ratio (HR) value. Additionally, statistical analysis was performed only on outcomes that were clear from the literature, and the ORR was analyzed by the odds ratio results. The statistical analysis was conducted using RevMan 5.4.1 software.

## 3. Results

### 3.1. Characteristics of Included Studies

Figure 1 shows the flowchart for study selection. A total of 1675 articles were searched, 122 duplicate articles were removed, and 1553 articles were screened. Of those, 1467 articles were excluded by title and abstract contents, and the full texts of 86 articles were evaluated. Unrelated articles such as reviews, protocols, and letters were excluded, and finally, six articles [32,33,34,35,36,37] were included (a total of 1410 patients). Table 2 shows the characteristics of the six articles included. The median ages of the participants were 49–60 years old (alectinib) and 51–61 years (crizotinib). Alectinib was administered at 600 mg in five trials [32,34,35,36,37]. However, in one trial [29], 300 mg was administered according to the approved dose of alectinib in Japan. In five trials [32,33,34,36,37], crizotinib was administered at 250 mg and in one trial, the patients were pretreated with crizotinib before receiving chemotherapy (pemetrexed 500 mg or docetaxel 75 mg) [35]. For the CDx test, VENTANA ALK immunohistochemistry assay was used in five trials [32,34,35,36,37], and Histofine ALK iAEP kit was used in one trial [33]. In all trials, PFS and ORR were presented as outcomes.

### 3.2. Quality Assessment

The results from the risk of bias assessment are presented in Table 3, and details are in the Appendix A. It was evaluated as high quality in two studies, and the quality of the other studies was appropriate. Of the papers evaluated as acceptable, 2/4 (50%) did not report OS results. In particular, 1/4 (25%) reported OS results but not HR results. The other 1/4 (25%) only reported the results for PFS and did not report the results for mPFS.

### 3.3. Primary Outcome

PFS was reported as an outcome in all studies. Table 4 shows the results of comparing the PFS values of alectinib and crizotinib. In Peters et al., PFS was longer for alectinib (68.4 months, 95% CI: 61.0–75.9) than for crizotinib (48.7 months, 95% CI: 40.4–56.9). The literature reports also indicated that the median PFS of patients treated with alectinib was better than those treated with crizotinib. This indicates that alectinib had superior efficacy than that of crizotinib (HR range 0.15–0.47).

### 3.4. Secondary Outcomes

ORR and DOR were reported as outcomes in all studies. Table 5, Table 6 and Table 7 show the results of comparing the ORR, DOR, and OS values of alectinib and crizotinib. The ORR was higher for alectinib (mean 79.5%) than for crizotinib (mean 65.2%) in all studies. The odds ratio (OR) confirmed that alectinib demonstrated a lower risk than crizotinib (OR: 2.21, 95%CI: 1.46–3.36, *p* = 0.0002, *I*^2^ = 39%, Figure 2). The DOR was better for alectinib than for crizotinib in all studies, and the HR (range 0.22–0.36) values also confirmed that alectinib was superior. OS results were not reported in most studies, but an analysis of HR values (range 0.28–0.89) confirmed that alectinib was associated with a lower risk.

## 4. Discussion

Personalized therapy focuses on developing and validating specific diagnostic tests that enable the identification of patients most likely to respond to a given treatment [28]. The ultimate goal, especially in oncology, is to utilize a molecular understanding of disease to optimize treatment for individual patients and to direct appropriate therapeutics to prevent disease. Therefore, personalized treatment in oncology requires a variety of studies to achieve the following goals: selecting the optimal therapeutic agent and dose, selecting the patient with the most effective response to a specific therapeutic agent with the greatest potential for side effect prediction, selecting and monitoring patients for efficient clinical trials, ensuring cost-effectiveness and therapeutic value in drug development, and working towards effective health care delivery and improvement [25]. Therefore, it is an important goal to generate sufficiently high-quality clinical evidence that can be utilized to determine an appropriate treatment strategy [38].

Precision medicine is rapidly shifting to a “targeted therapy and diagnostics” approach that works by acting on specific proteins [39]. Accordingly, the importance of CDx-using tests for various biomarkers is increasing. A CDx helps identify patients who will respond most positively to a specific treatment based on a diagnosis of molecular targets [3]. Developed targeted therapies are effective for patients with specific gene mutations or biomarkers and have fewer side effects [40]. In particular, several clinical studies have demonstrated that targeted therapy provides higher response rates and longer progression-free survival and overall survival [41]. However, validation that the IHC method for *ALK* rearrangement screening identifies clinical responses to target drugs is still lacking [42].

The purpose of this systematic review is to assess the clinical applicability of a companion diagnostic test (IHC method) by comparing the effects of alectinib and crizotinib in patients with *ALK*-positive NSCLC. Previous studies evaluated the diagnostic accuracy [29,30] and clinical value [30] of the IHC test in *ALK*-positive NSCLC patients. However, to the best of our knowledge, no systematic review has been reported on whether the IHC test has clinical applicability to the selection of patients suitable for target therapy. We hypothesized that alectinib would have a superior effect than crizotinib. The main highlight results can be summarized as follows: PFS was longer with alectinib (68.4 months, 95% CI: 61.0–75.9) than with crizotinib (48.7 months, 95% CI: 40.4–56.9) by 19.7 months. The ORR was higher with alectinib (mean 79.5%) than with crizotinib (mean 65.2%). The DOR and OS were confirmed by the HR value (<1), showing that alectinib had a lower risk than crizotinib. It is suggested that alectinib is more effective than crizotinib for the treatment of *ALK* mutations for several reasons as follows: Alectinib is a highly selective *ALK* inhibitor, can overcome secondary mutations in the kinase domain, and targets a different *ALK* tyrosine kinase domain than crizotinib [43]. In addition, alectinib can inhibit the effect of many *ALK* mutations (*EML4-ALK, F1156Y, G1269A, L1152R,* and *1151Tin*) identified in crizotinib-resistant patients [44]. In particular, the reason that the DOR could be prolonged in the study results seems to be because alectinib has a more powerful affinity for the *ALK* tyrosine kinase domain. It can be said that the PFS also became longer as the response depth of the DOR was increased [35]. Through this study, it was confirmed that the clinical applicability of companion diagnosis (IHC) is related to improvement in the efficacy of therapeutic agents.

In Camidge et al., biomarker evaluable population (BEP) subgroups were analyzed according to genetic variants (*EML4-ALK* 1-3) and types of samples (tumor and plasma) [32]. For PFS, alectinib was longer than crizotinib based on both tissue and plasma, and HR values (Tissue: 0.42 and Plasma: 0.32) were also significant (<1). This result suggests that for patients for whom tissue samples cannot be used in clinical settings, diagnosis using plasma samples may be possible. IHC analysis of histological samples is sensitive and highly specific, although germline variations may be somatic. However, when a new biopsy is required, it takes time to collect a sample and there is a burden incurred from repeated sampling due to tissue degeneration. Plasma samples are usually collected in tubes, which is very convenient and enables quicker collection. However, in limited samples, circulating tumor DNA (ctDNA) may not be detected, which may lead to low sensitivity and false positive results [45]. In Hida et al., a lower dose was used according to the approved dose of alectinib in Japan. This study demonstrated the superiority of alectinib over crizotinib with favorable safety evaluation results. Therefore, it was suggested that treatment with low doses of alectinib is also effective [33]. In Mok et al., the data were analyzed in three groups: (a) IHC (+) and FISH (+), (b) IHC (+) and FISH (uninformative), and (c) IHC (+) and FISH (−). In all three groups, alectinib showed better results than crizotinib. Therefore, a positive IHC result suggests that alectinib treatment can be considered even when the standard FISH test result is negative or uninformative [34]. In Novello et al., alectinib and chemotherapy in patients treated with crizotinib were compared. The results showed a good efficacy and safety profile with alectinib, suggesting that it can be considered a standard treatment, especially for extracranial and intracranial diseases [35]. Peters et al. compared crizotinib and alectinib in patients with previously untreated, advanced *ALK*-positive NSCLC, including asymptomatic central nervous system (CNS) symptoms. The results showed that alectinib was significantly more effective and confirmed that alectinib was effective in treating *ALK*-positive diseases in patients both with and without CNS disease [36]. In Zhou et al., crizotinib and alectinib were compared in Asian patients with *ALK*-positive NSCLC. The results were superior to alectinib and provided additional evidence that it was effective in Asian patients [37].

Currently, methods for detecting *ALK* rearrangement include FISH, IHC, RT-PCR, and NGS. Although FISH is a clinically validated reference standard test, it has limitations [11]. It requires advanced techniques such as fluorescence microscopy and is not cost-effective. In addition, *ALK* rearrangements are rare, making them difficult to apply to large-sized samples. RT-PCR is fast, simple, inexpensive and has a high-throughput. It is also applicable to non-tumor cells as only 10% tumor content is sufficient. However, careful pretreatment is required to obtain high-quality RNA, and the demanding nature of this may limit its clinical application [46].

NGS can simultaneously identify multiple driver genes by processing many genome manipulations in parallel to obtain a large amount of nucleotide sequence information, unlike IHC, which is based on antigen−antibody responses to specific protein molecules. In addition, it is possible to detect unbiased variants of cancer-associated genes in general on a target platform of cancer-associated genes. Targeted NGS is economically feasible for limited tissue amounts and other types of specimens (exudate, cytology, cell-free DNA, and urine.) compared to serial genetic testing. Thus, it may provide new opportunities for diagnosing and monitoring patients using liquid biopsy applications. However, relatively high-quality samples are required and data analysis can take a long time. The impact of NGS diagnosis will be greater as the proportion of novel molecular targets increases, as well as the ability to detect more variant types [9,47,48].

Because IHC can potentially detect overexpressed *ALK* chimeric proteins, it can routinely diagnose *ALK*-positive tumors and is a recommended pre-screening tool [49]. However, there is a limitation in that it is not possible to distinguish the exact expression of the *ALK* protein. Therefore, it may be required to perform additional confirmatory tests [50]. IHC, an antibody-dependent method, has been successfully applied for the analysis of anaplastic large cell lymphoma and inflammatory myofibroblast tumors [51,52]. The CDx devices reviewed in this study include the ‘Histofine ALK iAEP kit’ [30] and the ‘VENTANA ALK immunohistochemistry assay’ [32,34,35,36,37], both of which were approved as devices for screening patients receiving alectinib. The ‘Histofine ALK iAEP kit’, which is detected by the 5A4 antibody, uses a peroxidase reagent to attach many enzymes to the antigen site, and this is a more sensitive device than the conventional IHC method [53]. Savic et al. showed that the 5A4 IHC analysis showed more than 90% performanceand had the same as the D5F3 analysis [54]. The ‘VENTANA ALK immunohistochemistry assay’ uses the D5F3 antibody to detect the kinase region active in *ALK* expression using a high-sensitivity detection system to amplify *ALK*-specific signals [11]. The D5F3 antibody recognizes the C-terminal domain of *ALK* kinase in all pathological products, including complex *ALK* rearrangements not detected by FISH. Minca et al. showed excellent performance (sensitivity 94%, specificity 100%) of the IHC test using the D5F3 antibody [55]. As such, IHC may be a suitable tool for *ALK* rearrangement screening. Therefore, companion diagnostic testing using the IHC method is expected to be utilized in many clinical settings by reducing the exacerbation rate of NSCLC and identifying the appropriate treatment for each patient.

This study has a limitation in that the number of studies included is small. Therefore, additional research is needed to secure sufficient literature in the future. In addition, this study analyzed the clinical usefulness of CDx using IHC. As molecular diagnostics advance, a comparison of accuracy between next-generation analyses, such as NGS, is expected to become important.

## 5. Conclusions

This systematic review evaluated the clinical applicability of a companion diagnostic test (IHC method) by comparing the effects of alectinib and crizotinib in patients with *ALK*-positive NSCLC. In conclusion, alectinib demonstrated a longer progression-free survival, duration of response, and overall survival in patients with *ALK*-positive NSCLC. Also, alectinib had a higher objective response rate than crizotinib. Through this study, it was confirmed that the clinical applicability of companion diagnosis (IHC) is related to an improvement in the efficacy of therapeutic agents. Therefore, the companion diagnostic test (IHC method) is a clinically useful method for selecting alectinib in *ALK*-positive NSCLC patients.

## Figures and Tables

**Figure 1 diagnostics-12-01297-f001:**
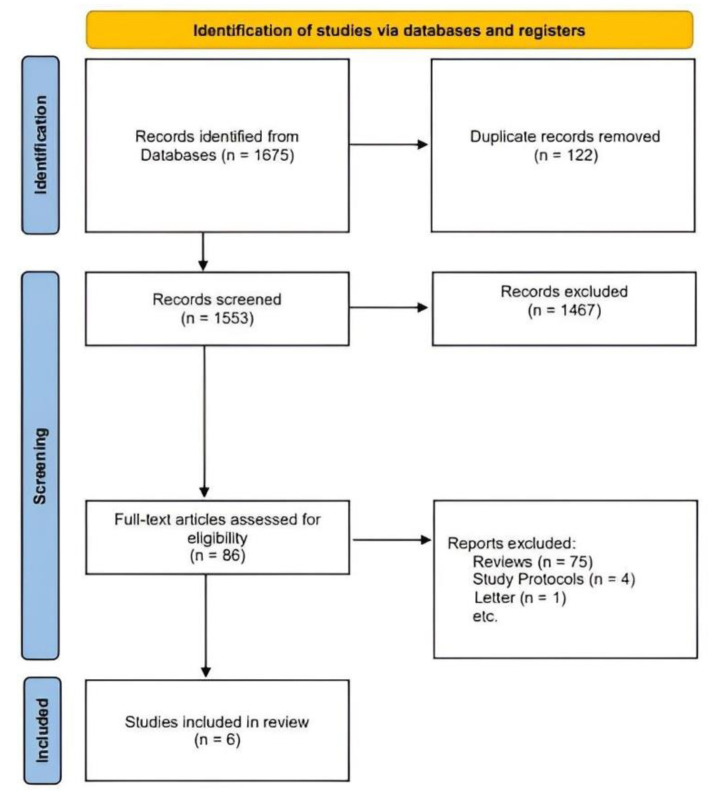
Flowchart of study selection.

**Figure 2 diagnostics-12-01297-f002:**
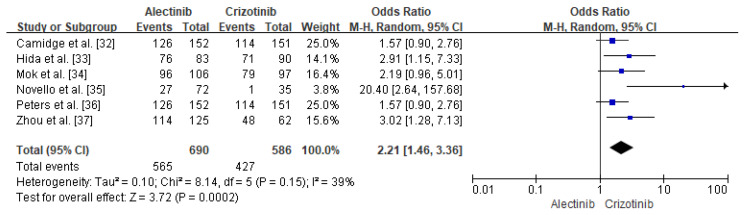
Forest plot of the ORR. The odds ratio of alectinib vs. crizotinib.

**Table 1 diagnostics-12-01297-t001:** Comparison of IHC and NGS analysis methods.

	IHC	NGS
Description	Antigen-antibody response to tissue, membrane, cytoplasmic, and nuclear molecules (antigens) that detect more than one specific protein molecule	Massively parallel sequencing of all DNA fragments present in a sample
Advantages	1. Application of the sample to fixed tissue (FFPE) 2. The most automated test in the area of biomarker evaluation 3. Excellent sensitivity for new antibodies (D5F3 83–100%, 5A4 93–100%) 4. Low cost	1. Can identify targetable molecular anomalies 2. Capable of detecting many molecular anomalies in a short time 3. Excellent detection sensitivity (85–100%)
Disadvantage	Additional testing required due to limitations in discriminating precise expression of *ALK* protein	1. Expensive cost 2. Increased complexity of molecular profiles in cases of abnormal mutations

**Table 2 diagnostics-12-01297-t002:** Characteristics of included studies.

No.	Study	Year	Study Type	Patients (*n*)	Age (A/C)	Intervention	Comparison	CDx Test	Outcomes
1	Camidge et al. [32]	2019	RCT	303	60	Alectinib (600 mg)	Crizotinib (250 mg)	VENTANA ALK immunohistochemistry assay	PFS, ORR, DOR
2	Hida et al. [33]	2017	RCT	207	61/59.5	Alectinib (300 mg)	Crizotinib (250 mg)	Histofine ALK iAEP kit	PFS, ORR, DOR, OS
3	Mok et al. [34]	2020	RCT	303	56.3/53.8	Alectinib (600 mg)	Crizotinib (250 mg)	VENTANA ALK immunohistochemistry assay	PFS, ORR, DOR
4	Novello et al. [35]	2018	RCT	107	55.5/59	Alectinib (600 mg)	Crizotinib + Chemotherapy	VENTANA ALK immunohistochemistry assay	PFS, ORR, DOR, OS
5	Peters et al. [36]	2017	RCT	303	56.3/53.8	Alectinib (600 mg)	Crizotinib (250 mg)	VENTANA ALK immunohistochemistry assay	PFS, ORR, DOR, OS
6	Zhou et al. [37]	2019	RCT	187	51/49	Alectinib (600 mg)	Crizotinib (250 mg)	VENTANA ALK immunohistochemistry assay	PFS, ORR, DOR, OS

Abbreviation: RCT: Randomized Controlled Trials; A/C: Alectinib/Crizotinib; CDx: Companion Diagnostics; ALK: Anaplastic Lymphoma receptor tyrosine Kinase; PFS: Progression-Free Survival; ORR: Objective Response Rate; DOR: Duration of Response; OS: Overall Survival.

**Table 3 diagnostics-12-01297-t003:** Quality assessment.

Study	Score
Camidge et al. [32]	++
Hida et al. [33]	++
Mok et al. [34]	++
Novello et al. [35]	+++
Peters et al. [36]	++
Zhou et al. [37]	+++

High Quality +++, Acceptable ++.

**Table 4 diagnostics-12-01297-t004:** Progression-Free Survival. (PFS; month).

No.	Study	Alectinib (95% CI)	Crizotinib (95% CI)	Hazard Ratio (95% CI)
1	Camidge et al. [32]	34.8 (17.7-NE)	10.9 (9.1–12.9)	0.43 (0.32–0.58)
2	Hida et al. [33]	NE (20.3-NE)	10.2 (8.2–12.0)	0.34 (99.7% CI 0.17–0.71)
3	Mok et al. [34]	34.8 (27.8-NE)	12.6 (9.1–14.8)	0.37 (0.25–0.56)
4	Novello et al. [35]	9.6 (6.9–12.2)	1.4 (1.3–1.6)	0.15 (0.08–0.29)
5	Peters et al. [36]	68.4 (61.0–75.9)	48.7 (40.4–56.9)	0.47 (0.34–0.65)
6	Zhou et al. [37]	NE (20.3-NE)	11.1 (9.1–13.0)	0.22 (0.13–0.38)

Abbreviation: CI: Confidence Interval; NE: Not Estimable.

**Table 5 diagnostics-12-01297-t005:** Objective Response Rate. (ORR; %).

No.	Study	Alectinib (95% CI)	Crizotinib (95% CI)
1	Camidge et al. [32]	82.9 (75.95–88.51)	75.5 (67.84–82.12)
2	Hida et al. [33]	92 (85.6–97.5)	79 (70.5–87.3)
3	Mok et al. [34]	90.6	81.4
4	Novello et al. [35]	37.5	2.9
5	Peters et al. [36]	82.9 (76.0–88.5)	75.5 (67.8–82.1)
6	Zhou et al. [37]	91	77

Abbreviation: CI: Confidence Interval.

**Table 6 diagnostics-12-01297-t006:** Duration of Response. (DOR; month).

No.	Study	Alectinib (95% CI)	Crizotinib (95% CI)	Hazard Ratio (95% CI)
1	Camidge et al. [32]	33.1 (31.3-NE)	11.1 (7.5–13.0)	-
2	Hida et al. [33]	NE (NE-NE)	11.1 (7.5–13.1)	0.32 (0.17–0.6)
3	Mok et al. [34]	33.1 (31.3-NE)	11.1 (7.4–14.7)	0.34 (0.22–0.53)
4	Novello et al. [35]	9.3 (6.9-NE)	2.7 (NE)	-
5	Peters et al. [36]	NE	11.1 (7.9–13.0)	0.36 (0.24–0.53)
6	Zhou et al. [37]	NE (18.4-NE)	9.3 (7.4-NE)	0.22 (0.12–0.40)

Abbreviation: CI: Confidence Interval; NE: Not Estimable.

**Table 7 diagnostics-12-01297-t007:** Overall Survival. (OS; %).

No.	Study	Alectinib	Crizotinib	Hazard Ratio (95% CI)
1	Camidge et al. [32]		-	-
2	Hida et al. [33]	7	2	-
3	Mok et al. [34]	-	-	-
4	Novello et al. [35]	22	20	0.89 (0.35–2.24)
5	Peters et al. [36]	-	-	0.76 (0.48–1.20)
6	Zhou et al. [37]	-	-	0.28 (0.12–0.68)

Abbreviation: CI: Confidence Interval.

## Data Availability

Data are available from the studies included in the review that have been cited.

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
