# Peer review of "A Systematic Review of Companion Diagnostic Tests by Immunohistochemistry for the Screening of Alectinib-Treated Patients in ALK-Positive Non-Small Cell Lung Cancer"

_diagnostics, 2022, doi:10.3390/diagnostics12051297_

Round 1

Reviewer 1 Report

Recently, I was invited to review an interesting paper entitled: A systematic review of companion diagnostic tests by immunohistochemistry for the screening of alectinib treated patients in ALK-positive non-small cell lung cancer. The paper is well written and easy to understand. I appreciate the interest of the authors in assessing the efficacy of IHC testing in ALK mutated NSCLC. It is important to discuss the accuracy of basic tests especially if that can influence the translation to the treatment effects. I have one major comment on the study.

I understand that it can be a good idea to highlight the accuracy of IHC staining by presenting the results of the treatment of different generations of ALK inhibitors. However, this approach contains a mental shortcut. A direct comparison of IHC and NGS would be significant support for the hypothesis. I would like to request the authors to compare the NGS and IHC in the form of a short table and add the section in the discussion.

Author Response

Manuscript ID: Diagnostics - 1689957

We would like to thank you very much for taking the time to review this study, and thank you for sharing the opinions of reviewers with good knowledge.

[Responses to the comments of Reviewer #1]

Authors: We would like to thank you for taking the time to review this study. In addition, thank you for your valuable opinion to make the article more complete.

Reviewer comment:

I would like to request the authors to compare the NGS and IHC in the form of a short table and add the section in the discussion.

Author Response:

  • The author supplemented the contents by adding a short table comparing IHC and NGS to the introduction (Line 76-80 and Table 1), and a section to the discussion (Paragraph 6, Line 300-310).

  • The authors noted the need for a study to compare the accuracy of IHC and NGS, as molecular diagnostics advance (Paragraph 8, Line 338-341).

Again, thank you for giving us the opportunity to strengthen our manuscript with your valuable comments and queries. We have worked hard to incorporate your feedback and hope that these revisions persuade you to accept our submission.

Reviewer 2 Report

The manuscript entitled:" A systematic review of companion diagnostic tests by immunohistochemistry for the screening of alectinib treated patients in
ALK-positive non-small cell lung cancer" focused on a systemic revision of literature daat about the role of companion diagnostic IHC- based tests in the deetction of ALK fusion transcripts is wel lwritten but require minor considerations to be accpeted for the publication:

  • In the table 2, please, could the authors explain the quality score for the analzyed study?
  • In the text, the authors did not report the failed test percentage performed with this approach. In my opinion, this data may represent a plus for the study. In addition, i would also suggest to better describe the standard specimens available for IHC analysis in terms of neoplastic cell percentage, sample type.
  • In the discussion section, i would suggest to better define the role of NGS systems in the analysis of clinically relevant fusion transcripts in ALK genes. To date, several literature evidencies have showed the great attituted of NGS platforms in the up front analysis of aberrant transcripts respect conventional methodologies (IHC, FISH). Could the authors discuss this point?

Author Response

Manuscript ID: Diagnostics - 1689957

We would like to thank you very much for taking the time to review this study, and thank you for sharing the opinions of reviewers with good knowledge.

[Responses to the comments of Reviewer #2]

Authors: We would like to thank you for taking the time to review this study. In addition, thank you for your valuable opinion to make the article more complete.

Reviewer comments:

  1. In the table 2, please, could the authors explain the quality score for the analzyed study? In the text, the authors did not report the failed test percentage performed with this approach.

  1. In addition, I would also suggest to better describe the standard specimens available for IHC analysis in terms of neoplastic cell percentage, sample type.

  1. In the discussion section, I would suggest to better define the role of NGS systems in the analysis of clinically relevant fusion transcripts in ALK genes.

Author Response:

  • The authors further explained the quality scores of the studies analyzed in the quality assessment. In particular, the reasons for the documents evaluated as 'Acceptable' were written (Results 3.2. Line 170-173).

  • IHC analysis of tumor cell percentage was added in the Introduction and sample type was added in the introduction (Paragraph 3, Line 58-63), and discussion (Paragraph 4, Line 268-274).

  • The role of the NGS was defined by adding a section to the discussion (Paragraph 6, Line 300-310).

Again, thank you for giving us the opportunity to strengthen our manuscript with your valuable comments and queries. We have worked hard to incorporate your feedback and hope that these revisions persuade you to accept our submission.

Reviewer 3 Report

The manuscript entitled "A systematic review of companion diagnostic tests by immunohistochemistry for the screening of alectinib treated patients in
ALK-positive non-small cell lung cancer" highlighted that the companion diagnostic test (Immunohistochemistry) is an effective test to determine whether to administer alectinib to ALK-positive NSCLC patients.

  • The Authors should discuss the role of immunocytochemistry (ICC) on cytological samples in ALK testing for NSCLC patients.
  • The Authors should provide the expand forms for all acronyms, including gene acronyms, through the text when they first appear.
  • Gene acronyms should be written in italics.

Author Response

Manuscript ID: Diagnostics - 1689957

We would like to thank you very much for taking the time to review this study, and thank you for sharing the opinions of reviewers with good knowledge.

[Responses to the comments of Reviewer #3]

Authors: We would like to thank you for taking the time to review this study. In addition, thank you for your valuable opinion to make the article more complete.

Reviewer comments:

  1. The Authors should discuss the role of immunocytochemistry (ICC) on cytological samples in ALK testing for NSCLC patients.

  1. The Authors should provide the expand forms for all acronyms, including gene acronyms, through the text when they first appear. Gene acronyms should be written in italics.

Author Response:

  • The authors add a description of the definition and role of immunocytochemistry (ICC) for cytological samples in the introduction (Paragraph 3, Line 60-69).

  • The authors corrected the gene abbreviation in italics and corrected the first appearing abbreviation in expanded form (Introduction, Paragraph 1).

Again, thank you for giving us the opportunity to strengthen our manuscript with your valuable comments and queries. We have worked hard to incorporate your feedback and hope that these revisions persuade you to accept our submission.
